# A Low-Cost, 3D-Printed Biosensor for Rapid Detection of *Escherichia coli*

**DOI:** 10.3390/s22062382

**Published:** 2022-03-19

**Authors:** Samir Malhotra, Dang Song Pham, Michael P. H. Lau, Anh H. Nguyen, Hung Cao

**Affiliations:** 1Department of Biomedical Engineering, University of California, Irvine, CA 92697, USA; smalhot2@uci.edu (S.M.); dspham1@uci.edu (D.S.P.); 2Sensoriis, Inc., Edmonds, WA 98026, USA; mlau@sensoriis.com; 3Department of Electrical Engineering, University of California, Irvine, CA 92697, USA

**Keywords:** biosensor, cyclic voltammetry (CV), electrochemistry, electrode, *Escherichia coli* (*E. coli*), ferrocene, thyroxine

## Abstract

Detection of bacterial pathogens is significant in the fields of food safety, medicine, and public health, just to name a few. If bacterial pathogens are not properly identified and treated promptly, they can lead to morbidity and mortality, also possibly contribute to antimicrobial resistance. Current bacterial detection methodologies rely solely on laboratory-based techniques, which are limited by long turnaround detection times, expensive costs, and risks of inadequate accuracy; also, the work requires trained specialists. Here, we describe a cost-effective and portable 3D-printed electrochemical biosensor that facilitates rapid detection of certain *Escherichia coli* (*E. coli*) strains (DH5α, BL21, TOP10, and JM109) within 15 min using 500 μL of sample, and costs only USD 2.50 per test. The sensor displayed an excellent limit of detection (LOD) of 53 cfu, limit of quantification (LOQ) of 270 cfu, and showed cross-reactivity with strains BL21 and JM109 due to shared epitopes. This advantageous diagnostic device is a strong candidate for frequent testing at point of care; it also has application in various fields and industries where pathogen detection is of interest.

## 1. Introduction

*Escherichia coli* (*E. coli*)—a Gram-negative bacillus—is prevalent in the gastrointestinal tract of humans and warm-blooded animals, and well-known as one of the most significant pathogens [1]. Even though deemed harmless, numerous *E. coli* strains have acquired specific virulence attributes, making them pathogenic for humans and animals. The combination of virulence factors has continued to become specific pathotypes, such as enteropathogenic *E. coli* (EPEC), Shiga toxin-producing *E. coli* (STEC), enteroaggregative *E. coli* (EAEC), enterotoxigenic *E. coli* (ETEC), enteroinvasive *E. coli* (EIEC), and diffusely adherent *E. coli* (DAEC), have caused various infectious diseases in living things [2].

Current laboratory-based methods to detect food-borne pathogens depend on biochemical or microbiological tests, which have long turnaround times, expensive costs, and/or inadequate sensitivity and specificity [3]. Techniques used to analyze microbiological specimens are bacterial culturing, cytometry, immunoassays, microscopy, and PCR, among others. Therefore, there is an urgent demand for a rapid and cost-effective test that is capable of identifying whole-cell bacteria with adequate sensitivity and specificity. Here, we propose a 3D-printed, host-cell detection biosensor for *E. coli* using a modified graphite pencil as the sensing electrode. This sensing platform has been used in many kinds of biosensor engineering [4]. For instance, De Lima et al. reported using a low-cost pencil graphite electrode coated by gold nanoparticles to detect SARS-CoV-2 on a minute scale [5]. In this study, we used thyroxine, a dual functional group molecule, and ferrocene to modify the graphite pencil electrode. This modification allows antibody conjugation and increases electrode transfer rates. Cross-reactivity studies of five different *E. coli* strains (DH5α, BL21, TOP10, and JM109) were performed to investigate the specificity of the biosensor. Due to its effective methodology, rapid diagnosis, and reduced costs, our biosensor can be a potential candidate for high-frequency testing at point of care as well as applicable to multiple fields where pathogen detection is of interest.

## 2. Materials and Methods

### 2.1. Materials

1-Ethyl-3-(3-dimethyl aminopropyl) carbodiimide (EDC) (98%), N-Hydroxysuccinimide (NHS) (98%), L-Thyroxine (98%), Ferrocene (99%) were purchased from Thermo Scientific. Graphite pencil lead with 0.7 mm diameters was purchased from a local Walmart store. Anti-*E. coli* antibody (ab217910) was obtained from Abcam, UK. Four *E. coli*-competent strains, DH5α, BL21, TOP10, and JM109, were from Real Biotech Corporation (RBC) in Taiwan. Electrochemical measurements were carried out using a CH760E potentiostat, controlled by the CH Version 12.04 software. A 3D-printed chamber containing three electrodes was used as the electrochemical cell (0.5-mL volume). Graphite pencils (0.7 mm) were used as working, reference, and counter electrodes. The graphite lead that was used as the reference electrode was coated with Ag/AgCl paste. All reagents and the deionized water (resistivity ≥ 18 MΩ cm at 25 °C) used in this work were of analytical grade.

### 2.2. Electrochemical Characterization

Cyclic voltammetry (CV) and electrochemical impedance spectroscopy (EIS) were used to measure the electrochemical properties of the electrodes in each modification step, according to de Lima et al. (2021) [5]. A potential window from 0.7 to −0.5 V with a scan rate of 50 mV·s^−1^ was set up for CV experiments. In square wave voltammetry (SWV), potentials were scanned from −0.8 to 0.4 V, corresponding to a frequency of 60 Hz, amplitude of 65 mV, and step low of 6 mV. In EIS, the frequency ranges from 0.1 Hz to 1 × 10^6^ Hz using an amplitude of 10 mV, and under an open circuit potential, the potential was conducted. A 0.1 mM KCL solution containing 5.0 mM of the mixture [Fe(CN)6]^3−/4−^ solution was used as an electrolyte for all electrochemical characterization. Morphological characterizations of GPE before and after superficial functionalization were performed using the Tescan GAIA3 SEM-FIB, a specific scanning electron microscopy (SEM), from the University of California, Irvine Material Research Institute (UC IMRI). SEM images were recorded with 129 to 71,000 times magnifications, acceleration voltage of 30 kV, and using the In-Beam SI mode. All electrochemical experiments were carried out at room temperature (25 ± 3 °C).

### 2.3. Modification of Graphite Electrodes with Ferrocene

The working electrode (WE) was polished with a 2000-grit sandpaper, and a working area of 1.5-cm length by 0.7-mm diameter was obtained. The GPE was immersed in 10 mM L-Thyroxine dissolved in dimethylsulfoxide (DMSO) for 2.0 h as a first modification step. This process allowed the graphite surface to be functionalized with the iodine groups. Then, the modified graphite substrate was kept in a 5 mM Ferrocene solution in toluene, which enabled ionization anchoring of Ferrocene to the iodine groups of thyroxine on the electrodes. The modified electrode was washed three times in a 10 mM PBS buffer.

Next, the electrodes were incubated in the solution containing 100 mM EDC and 50 mM NHS diluted in 0.1 M MES (pH = 5.0). Subsequently, 0.3 mg. mL^−1^ of anti-*E. coli* antibody was added into the tube containing the activated electrodes and vortexed for 5 s. After 10 min, the antibody was immobilized onto the substrate, reached the anchoring stability, and provided a highly sensitive SWV response. In the presence of EDC and NHS, the activated groups of thyroxine covalently bonded to the primary amine at the N-terminal (or side-chain amine groups) of antibodies. The reaction between the carboxyl groups and EDC–NHS resulted in the formation of a stable ester, which undergoes nucleophilic substitution with the amine groups present on the modified electrode surface. This resulted in the formation of an amide bond between the modified GPE/Ferrocene/Thyroxine surface and the antibodies. In the final step, nonspecific sites present within the electrode surface were blocked by incubation in a 1% (mass/vol) BSA solution for 30 min.

### 2.4. Bacterial Pathogen Sensing Using the Developing Sensor

For diagnosing bacterial cells, a volume of 500 μL of each bacterial strain was applied to the three-electrode configured sensing system (Figure 1) and incubated for 5 min. After the incubation period, the sensing system was gently washed with 0.1 mM PBS buffer (pH 7.4) to remove unbound bacterial cells. Next, 500 μL of redox probe solution (5.0 mM [Fe(CN)6]^3−/4−^ in 0.1 M KCl) was injected into the chamber for voltammetry measurements and detection of current decrease due to binding of bacterial cells to the biosensor. Subsequently, the electrochemical response was monitored using the SWV technique. The specificity and cross-reactivity of the biosensor were evaluated in spiked samples. The total diagnostic time was calculated to be 13 min, which considered the sample incubation period (10 min), the time required to record two SWVs (before and after sample incubation, 2 min), and the washing step after sample incubation (1 min).

The limit of detection (LOD) and limit of quantification (LOQ) of the sensor were calculated according to the four-parameter logistic curve, using Equations (1) and (2):
L_B_ = μ_blank_ + t_(1−α, n−1)_ × σ_blank_(1)
L_S_ = L_B_ + t_(1−β, m (n−1))_ × σ_test_(2)
where L_B_ is a value of blank limit and L_S_ is the LOD in the signal of samples; μ_blank_ is the mean of signal intensities for n blank replicates; σ_blank_ is the standard deviation of blank replicates; σ_test_ is the pooled standard deviation of n test replicates; t_(1−α, n−1)_ is the 1 − α percentile of the t-distribution given n − 1 degree of freedom, referring to the number of independent blank measurements; and t_(1−β, m (n−1))_ is the 1 − β percentile of the t-distribution given m (n − 1) degrees of freedom, referring to the number of independent sample measurements, both at the 95% confidence interval [6].

### 2.5. Specificity and Cross-Reactivity Studies

The cross-reactivity study was performed by recording the analytical signal (current decrease of the redox probe) obtained from different batches with 1000 bacterial cells on three times measurements. The specificity of the sensors was evaluated by the analytical sensitivity value extracted from the analytical curves in a concentration range from 25 to 6400 cells stored in PBS buffer (pH 7.4) and LB media. The specificity studies were carried out using the five different *E. coli*-competent strains, each at 100 colony-forming units (cfu). The anti-*E. coli* antibody was used to evaluate the capability of the sensor to detect DH5α.

The performance of the biosensor was assessed using simulated samples prepared by mixing five strains in culture media. We set a current decrease cutoff value lower than 55 μA for diagnostic purposes in accordance with the analytical response obtained for the lowest concentration of bacterial cells detected (150 cfu) in the cell–response curve. Samples that exhibited the cut-off value over 15 µA were identified positive for DH5α. All sample results were analyzed and compared to those obtained from quantitative polymerase chain reaction (qPCR). A total of 100 µL of 100 cells were added to 400 µL of tap water. The resulted 500 µL of each sample were centrifuged at 7000 rpm for 10 min to collect cells for colony qPCR with SYBR master kits (Thermo Fisher). The threshold cycle (Ct) values obtained by qPCR for spiked samples from 19.8 to 27.7 were considered as a presence of *E. coli* in the sample.

## 3. Results and Discussion

### 3.1. Electrochemical Biosensor Design

The electrochemical device was designed to measure the binding affinity between the anti-*E. coli* antibody and *E. coli* cells (Figure 1). The monoclonal antibody was used as a recognition element to ensure sensitive and selective bacteria detection. The graphite that was placed in a WE were functionalized by the drop-casting method with thyroxine and ferrocene before conjugation with the monoclonal antibody (Figure 1A,B). The graphite WE were polished with 2000-grit sandpaper to remove impurities from the surface, and a contact area of 1.5-cm length by 0.7-mm diameter was obtained. Next, to form cross-linked elements, the WE were immersed in a 10 mM L-Thyroxine in DMSO for 2.0 h as a first modification step. Thyroxine molecules have dual functional groups, such as iodine and amine groups, that facilitate the ionization and covalent attachment of ferrocene and antibodies containing carboxyl-terminal moieties. Here, we leveraged thyroxine to modify the GPE’s surface with ferrocene, a versatile redox molecule. Subsequently, the coupling reaction was conducted in two different steps. First, we added the modified electrodes in a 0.1 M MES solution containing the prepared reactive intermediary EDC and NHS to form reactive coupling linkers. This enables the anchoring between the amine groups of the thyroxine and carboxyl groups of antibodies, yielding Ab/thyroxine/ferrocene/GPE after 30 min of two-step activations at room temperature [6]. The electrodes were then incubated with bovine serum albumin (BSA) at 37 °C for 30 min to block the electrode’s unspecific binding sites after immobilization of Ab. BSA is a common blocking protein with a high density of positive lysine residues that are commonly used for blocking procedures. Next, we exposed the sensor to samples containing bacterial cells. The decrease in the peak current of a redox probe ([Fe(CN)6] ^−3/−4^) enabled the diagnosis of bacteria-free samples versus those that were infected with bacterial cells (Figure 1C). Our biosensor chip is shown in Figure 1D.
Figure 1Overview of the system. (**A**) Scheme of developing an electrochemical biosensor for whole-cell detection. (**B**) Thyroxine/Ferrocene functionalization on pencil graphite electrodes and antibody conjugation through ECD and NHS activation. (**C**) The concept of electrochemical response of the sensor in the presence of *E. coli* is based on the current drop due to the specific binding of DH5α to antibody-coated onto electrodes. The *E. coli* cells bound onto the electrode prevent electron transfers from the redox probe ([Fe(CN)6]^3−/4−^) to the working electrodes. (**D**) An electrochemical cell containing a three-electrode configuration was made from a 3D-printing protocol.
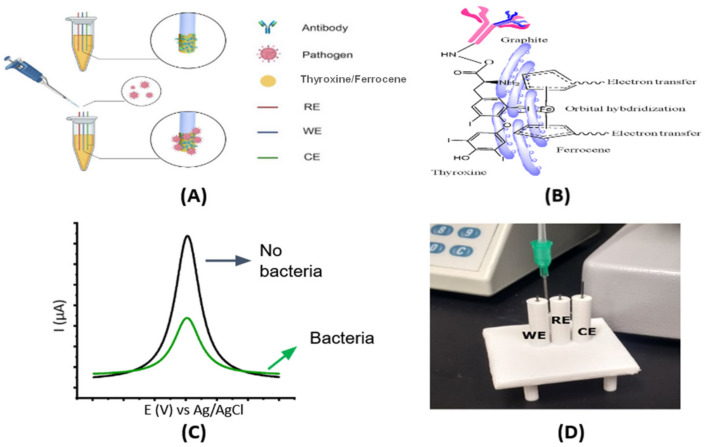


### 3.2. Characterization of the Biosensor

Electrochemical experiments were then performed to characterize the biosensor. The bare GPE presented a flat surface containing stacked carbon sheets (Figure 2A). Successful formation of the ferrocene and thyroxine complex was confirmed by cyclic voltammetry, presenting a specific redox/oxidative peak at −0.1 and 0.1 e/V vs. Ag/AgCl (Figure 2B,C). The thyroxine coated on the WE surface after the optimized functionalization process, facilitating the subsequent anti-*E. coli* antibody immobilization onto the surface of the electrode. The electrochemical behavior of each functionalization step (Figure 2C) was analyzed by CV and electrochemical impedance spectroscopy (EIS) (Figure 3A). The CV data and Bode plots revealed that the bare GPE electrode (brown line) possessed a resistance to charge transfer of 66.8 ± 8.7 Ω, indicating a small resistance toward redox conversion of the ([Fe(CN)6]^−3/−4^) complex and a high rate of electron transfer on the electrode surface. This result was in agreement with the current value of 135.3 ± 7.3 μA shown for the same electrode by the CV plot (brown line). Next, we modified the WE with thyroxine (red line), leading to increased current resistance to 73.1 ± 5.6 Ω and a decreased current of 120.8 ± 8.9 μA. These data indicate that thyroxine acts as an electrical insulator hindering the electron transfer at the interface of the WE by inhibiting the redox probe from reaching the WE surface. Ferrocene was then anchored covalently to the surface of the GPE (blue line) through an ion bond between the iodine group from the thyroxine and the ferric groups from the ferrocene. The functionalization of ferrocene modified GPE led to decreased values of current resistance (38.7 ± 1.3 Ω) and increased transfer current (180.7 ± 2.3 μA) compared to the previous functionalization step (Figure 2C and Figure 3A). Figure 3B shows a magnitude view of the plots in high-frequency regions. The higher current and lower charge transfer resistance detected resulted from the greater electrocatalytic and surface area presented by the ferrocene [7], which contributed to fast electron-transfer kinetics, thus conferring attractive features for sensor development [8]. In addition, the free amine, iodine, and ferric cation functional groups presented on the thyroxine/ferrocene–modified WE led to favorable electrostatic interactions of the anionic probe [Fe(CN)6]^3−/4−^, providing an accumulation of the redox probe surrounding to the electrode interface. This improved electrochemical response, such as higher current peak, especially for whole-cell detection. As a final functionalization step, we immobilized BSA to block the remaining unmodified electrochemical sites and to avoid nonspecific and undesired adsorption of other molecules (Figure 3C). This step resulted in the highest resistance values (305.6 ± 17.8 Ω) and lowest current (25.4 ± 6.5 μA), suggesting a continued decline in the charge transfer kinetics after anchoring BSA due to organic materials.

### 3.3. Analytical Biosensing of the Sensor

We used SWV for bacterial cell detection. This technique is highly sensitive, especially for detecting reversible redox species, such as potassium ferricyanide and ferrocyanide. The sensing approach relied on the current signal decrease induced by specific interactions between the antibody and *E. coli* cells. A greater pathogen concentration leads to a greater decrease in the current signal of the redox probe [Fe(CN)6]^−3/−4^. This means that bacteria bound to WE partially blocked the redox probe’s access to the WE surface. The instrumental parameters, including the amplitude, frequency, and step potential, were optimized to enhance the capacity detection of the *E. coli* cells. The highest peak current values for the redox probe were obtained using an amplitude potential of 90.0 mV, frequency of 115.0 Hz, and step potential of 10.0 mV. We obtained an analytical curve for different concentrations of *E. coli* in 100 mM (PBS) (pH = 7.4) under optimized experimental conditions. The experiments were recorded in triplicate using an increased number of *E. coli*, from 25 to 6400 cfu. The SWV signal obtained at each cell number was shown in Figure 4A. The linear regression curve was calculated at concentrations ranging from 25 to 6400 cfu of *E. coli*, resulting in an analytical sensitivity value of 55 ± 1.8 μA·cfu^−1^·mL^−1^ and a linear coefficient of 0.97 (Figure 4B). Next, we conducted the optimal incubation time for detecting *E. coli* in spiking samples by evaluating the binding sensitivity parameter obtained from the cell–response curves at 400 *E. coli* cells. The spiked samples were prepared by added approximate 100 cells to 400 μL of sample matrix of tap water and used 1% of BSA for an unspecific assay. Before loading onto the chip, the samples were vortexed well. The results were expressed as ΔI = I − I_0_, where I_0_ and I correspond to the current recorded for the redox probe ([Fe(CN)6]^3−/4−^) after and before incubating the sample, respectively. The optimal incubation time was determined approximately in 5 to 7 min due to the highest value of the angular coefficient of the cell–response curves, referring to binding kinetics between the antibody and *E. coli*. Note that the SWV response for the redox probe [Fe(CN)6]^3−/4−^ decreased with increased concentration of *E. coli* due to the decrease in the current signal, which induced a specific interaction between the *E. coli* and the antibody-coated WE (Figure 4C). The binding of *E. coli* to the biosensor surface partially increases nonconductive layers on the electrode, leading to a current drop and leading to indicate the presence of *E. coli* in the samples.

SWV is commonly used for assays that determine biological binding interactions and reflect the underlying binding kinetics even with small molecules [9]. The sensor enabled the rapid detection of *E. coli* cells (less than 100 cfu), providing high sensitivity, and a low LOD in a complete 3D-printing device with pencil graphite used as a WE. The threshold cycle (Ct) of the RT-PCR data (forward primers 5′-TCG ACC TGA TAT CCC TGT TGT TG-3′; reverse primer 5′-GTG TCA TCT AAA GGC TGC GTG-3′) for a cell concentration from 25 to 400 cells ranged from 21.7 to 29.3 Ct.

### 3.4. Cross-Reactivity Assays

Cross-reactivity studies between *E. coli* strains were carried out to investigate the specificity of our biosensor device toward DH5α and rule out potential off-target reactivity (Figure 5). Using the same experimental conditions as for DH5α (Figure 4), we tested four other *E. coli* strains, including, DH5α, TOP10, BL21, and JM109. While delta current drop (ΔI) of DH5α (Figure 5A) and TOP10 (Figure 5B) were −11.5 ± 1.2 and −9.7 ± 1.5 µA, respectively, the ΔI of BL21 (Figure 5C) and JM109 (Figure 5D) approximately reached zero. The results showed that the antibody that was used in this research showed cross-reactivity to TOP10. However, BL21 (Figure 5C) and JM109 (Figure 5D) showed less cross-reactivity to the coated antibody.

The 27 spiked samples were prepared for further evaluation of cross-reactive *E. coli* DH5alpha antibodies to the TOP10, BL21, and JM109 species. Figure 6A showed specificity with the antibody-coated into WE, which presented a current drop (ΔI) lower than the cutoff value of 7.5 μA obtained by SWV for the lowest *E. coli* concentration detected. The cross-reactivity could be due to shared epitopes of the surface proteins between the *E. coli* strains that led to a reduced specific binding affinity toward whole cell detection [10]. Shorter DNA fragments of TOP10 can be amplified by the primer pairs that are specific to DH5alpha (Figure 6B). However, the correlation analysis from 27 spiked sampled showed that the correlation in the robustness between the proposed biosensor and qPCR were low (R^2^ = 0.78). It is necessary to increase more samples and develop recognition receptors for whole-cell biosensors based on species-specific epitopes or aptamers, which should be a part of biosensor engineering.

## 4. Conclusions

We present a simple, inexpensive, and portable electrochemical biosensor based on a 3D-printing technique that enables the diagnosis of pathogenic bacteria within 15 min, requiring just 500 μL of sample, utilizing highly accessible and commercially available materials (i.e., graphite pencil leads and 3D printing). Each test costs only USD 2.50. The WE can be functionalized in less than 3 h and remains stable for over 5 days when stored in a PBS solution at 4 °C. The sensor displayed high sensitivity for detecting whole *E. coli* cells (53 cfu and 270 cfu) and showed cross-reactivity with TOP10 and JM109 due to shared epitopes. The robustness and accuracy of the sensors were successfully evaluated by analyzing with other methods for whole-cell detection (Table 1), indicating that our method can be an alternative to accurately detect *E. coli* in spiking samples, but it needs to be challenged in real samples. Additionally, it could enable monitoring other pathogens since the modification of the electrodes with other recognition elements can be readily performed; it is also easy to operate and could integrate into a wireless system. Finally, the sensor can be applied for detecting pathogens in the areas of biosafety, aquaculture, and water quality if the recognition elements (like aptamers and antibodies) are available.

## Figures and Tables

**Figure 2 sensors-22-02382-f002:**
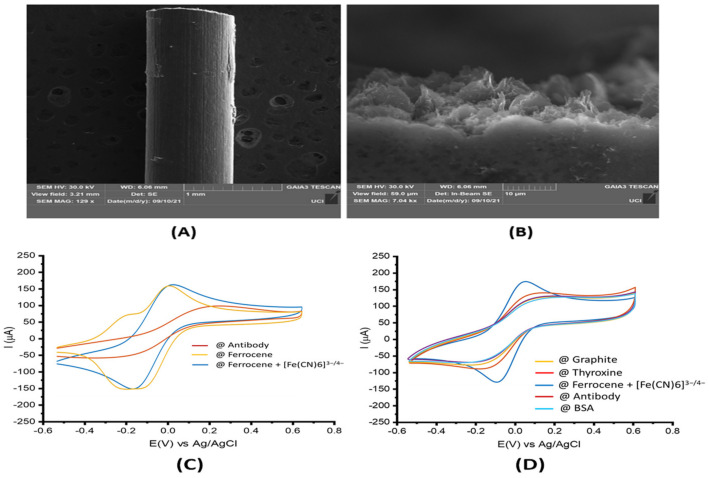
Characterization of the pencil graphite electrode. (**A**) SEM image of the bare 0.7 mm pencil graphite electrode surface at 129× magnification. (**B**) SEM image of the 0.7 mm pencil graphite electrode after functionalization with Thyroxine and Ferrocene at 7.04 kx magnification. (**C**) CV was recorded for each bioconjugation step of the working electrode in a solution of 5.0 mM [Fe(CN)6]^−3/−4^ containing 100 mM KCl as the supporting electrolyte at a scan rate of 50 mV·s^−1^. Ferrocene solution was prepared in acetonitrile with 0.1 M KCl electrolyte-supported solutions. (**D**) Electrochemical properties of the electrode surfaces after each step of the surface modification.

**Figure 3 sensors-22-02382-f003:**
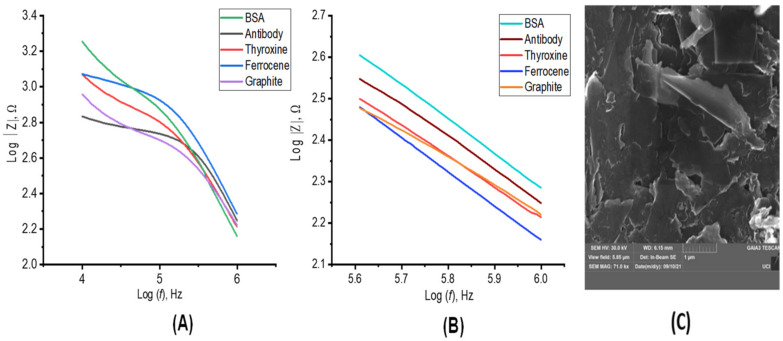
Characterization of pencil graphite electrode. (**A**) Bode plots were obtained using the same conditions as in 2C. (**B**) An amplitude view of the plots in high-frequency regions. (**C**) SEM image of the working electrode after being coated and blocked by antibody and BSA, accordingly, at 71.0 kx magnification.

**Figure 4 sensors-22-02382-f004:**
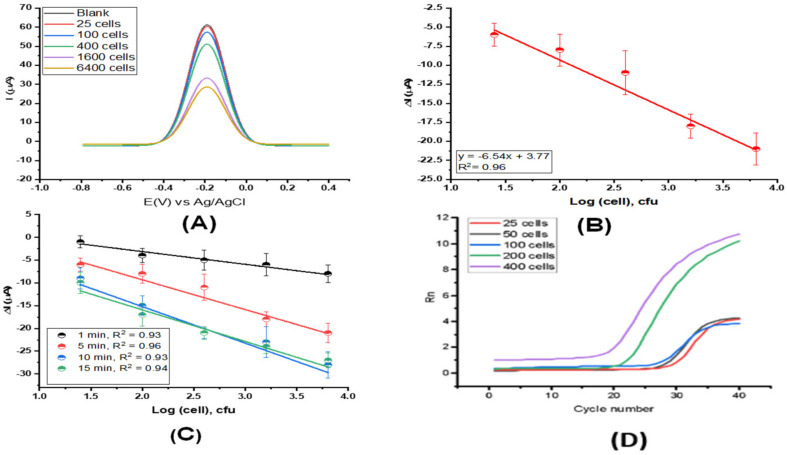
Sensing study of *E. coli* bacteria on the sensor. (**A**) Baseline-corrected SWVs of the 5.0 mM [Fe(CN)6] ^3−/4−^ redox probe containing 0.1 mol·L^−1^ KCl as the supporting electrolyte after incubating the working electrode with different concentrations of *E. coli* cells ranging from 25 to 6400 cells (approximately). (**B**) Linear regression (triplicate experiment) was retrieved from Figure 4A, using the drop of the current signal. (**C**) A time course (from 1 to 5 min) of binding pattern of *E. coli* cells on the modified electrode using 25 to 6400 *E. coli* cells. A distinct linear regression curve was obtained after 3 min. (**D**) Validation data of the presence of *E. coli* in the samples by qPCR.

**Figure 5 sensors-22-02382-f005:**
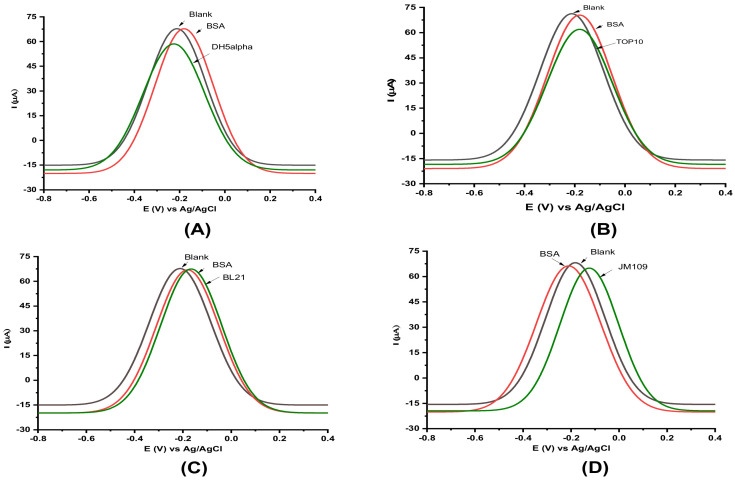
Specificity studies of the sensor using four laboratory *E. coli* strains. (**A**) DH5α, (**B**) TOP10, (**C**) BL21, and (**D**) JM109. Black lines are the baseline-corrected SWV before incubating with the bacteria sample. Red lines for 1% of the BSA solution as an unspecific binding. Green lines for each graph are the result after incubation with the specific *E. coli* strain. Experimental conditions were conducted as the same conditions described in this study.

**Figure 6 sensors-22-02382-f006:**
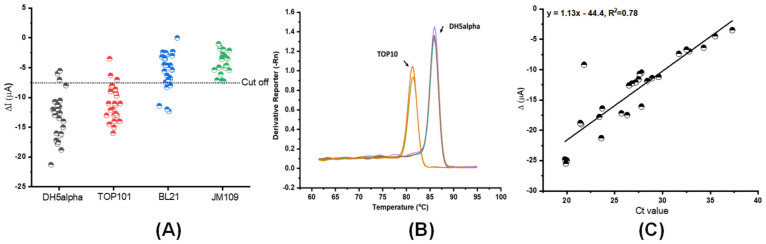
(**A**) Evaluation of cross-reactive *E. coli* DH5alpha antibodies to TOP10, BL21, and JM109 species of *E. coli*. The cut-off value was obtained from the Receiver Operating Characteristics (ROC) Curve (not shown). (**B**) Melting curves for specific amplification analysis. (**C**) Biosensing measurements of *E. coli* DH5alpha compared with colony qPCR by a regression procedure.

**Table 1 sensors-22-02382-t001:** Examples of the characteristics of conventional methods and biosensors for whole-cell detections.

Assays	Advantages	Disadvantages	Detection Signal	LOD (cfu.mL^−1^)
PCR	No enrichment steps Multiple detections	Time: 2–5 h No discrimination between viable and non-viable cells	UV light, Fluorescence	100–10^5^ [11,12]
qPCR	Real-time No post-amplification g Multiple detections Time: 1.5–2.5 h	Use of fluorescent tags	Fluorescence	1–100 [13,14]
Culture	Particular bacteria species	Excessively time-consuming Different selective media Time: 5–14 days	Phenotype Enzyme acitivities	1–100 [15]
Immunological Methods	A wide range of targets Time: 2–4 h	Cross-reactivity of antigens	Fluorescent chromogenic	30–1000 [16]
Optical biosensor	High selectivity and sensitivity Label free Time: 10–20 min	Fluorescent probes Complex system (SPR) Adds time and cost to the procedure.	Refractive indexes, Fluorescent, Light scattering	15–100 [17]
Square wave voltammetry biosensor	High selectivity and sensitivity Label free Time: 5–15 min	High probability of cross-sensitivity.	Electrical signal	10–1000 [18,19]
Impedimetric immunosensors	Non-destructive mechanism wide range of material, Time: 5–15 min	High probability of cross-sensitivity	Electrical signal	3–500 [20,21]

## Data Availability

Not applicable.

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
