# Peer review of "A Low-Cost, 3D-Printed Biosensor for Rapid Detection of Escherichia coli"

_sensors, 2022, doi:10.3390/s22062382_

Round 1

Reviewer 1 Report

Dear authors, 

The manuscript "A Low-Cost, 3D-Printed Biosensor for Rapid Detection of Escherichia coli"  is very interesting, especially as it features a low-cost and easy-to-use bacterial monitoring system. Furthermore, it is offered as a versatile system also for measurements to be carried out for the determination of many other high molecular weight and dangerous substances (bacteria, viruses, etc.). The only thing missing is its application on a real sample and a data on its stability over time after its preparation (it would be interesting to have it "ready to use").
  minor corrections
carefully check the data presented and the decimal places of the standard deviations 

Author Response

Review # 1. Comments and Suggestions for Authors

Comment. The manuscript "A Low-Cost, 3D-Printed Biosensor for Rapid Detection of Escherichia coli” is very interesting, especially as it features a low-cost and easy-to-use bacterial monitoring system. Furthermore, it is offered as a versatile system also for measurements to be carried out for the determination of many other high molecular weight and dangerous substances (bacteria, viruses, etc.). The only thing missing is its application on a real sample and a data on its stability over time after its preparation (it would be interesting to have it "ready to use"). Minor corrections carefully check the data presented and the decimal places of the standard deviations 

Author Reply. Thank you very much for your comments. We have a plan to upgrade and integrate the sensor into a wireless system developed by our lab to measure real samples of bacteria in fields. Data on its stability over time including batch-to-batch reproducibility, storage, shipping conditions, and chemical and physical parameters of real samples will be determined. Due to the limit of Laboratory and biosafety regulations, we only use spiking samples for this experiment to characterize and determine sensing parameters. We will take a note to add data of the stability of the sensors in the next study.

Regarding data presentation and decimal places of standard deviations, we improved the quality of data presentation and correct the decimal places of standard errors. Please refer to Figures 4A and C for your consideration.

Reviewer 2 Report

In the submitted manuscript, the development of an electrochemical immunosensor is reported for rapid and low-cost detection of bacteria.

The topics is of high interest but major corrections are needed before the manuscript can be considered for publication in Sensors journal. Detailed comments are detailed below.

Materials and methods:

Lines 89-90 (and fig. 1B) describe the activation of carboxylated electrode surface through EDC/NHS chemistry, followed by chemical binding of antibodies via their amine groups. Some lines after (line 94), and further in the manuscript (lines 159-160), the authors report that activated carboxyl groups of antibodies react with surface amine groups. EDC/NHS chemistry actually activate carboxyl groups which can react with amine groups, not the opposite. Please make the adequate corrections.

How are the numbers of bacterial cells determined? Are the different suspension concentrations obtained by successive dilutions of a mother suspension?

It is mentioned that the specificity studies are performed using the 5 strains at 100 cfu. Results presented in Figure 5A are not consistent with this concentration and Fig 4A.

Results and discussion

Fig.1B : electron transfer for ferrocene/ferrocenium couple is a one electron process that occurs via iron not benzene rings. Does the ferrocene-tyroxine complex electrochemistry rely on a two-electron process, as suggested in the figure? Please provide a suitable reference that describes the related mechanism. Are there still free iodine groups at the electrode surface (as suggested in the figure and in line 205), or are they all in the Fe complexed form? If any, they will generate repulsion effects (and not attractive interactions, line 206) towards the ferro/ferricyanide anionic probe.

Fig.2C: with the chosen representation (increasing current vs increasing potential), negative peaks (reduction) come on the left part of CV and positive peaks (oxidation) come on the right part of CV. Please correct the figure.

Why is Ferro/ferricyanide couple used for detection and not ferrocene- directly? This should be clearly explained. Signals obtained in absence of ferrocene and in presence of ferrocene but in absence of ferro/ferricyanide couple should be given for justification.

Fig.3A and B: The common Bode representation plots logZ vs Logf, not Z vs f. Please, correct also Y axis titles.

Fig.4A: what was the incubation time ? The blank signal (in absence of bacteria) should be also represented

Fig.4B: equation of the regression line given in the figure is not correct. How are the error bars calculated ? (how many replicates, how are they obtained?). This  curve  (calibration curve) should be represented as delta I vs loc C.

Fig.4C: please correct X-axis values and title as well as Y values. How do you explain the large decrease of slope at 2 min?

Line 249: "…approximately in 5 to 7 min". This is not consistent with results presented in Fig4C. The figure or the text should be modified

Table 1 is not useful, primers and PCR length can be provided in the materials and methods section, directly in the text.

line 262: it can not be affirmed that a regression coefficient of R2= 0.954 is the sign of high linearity.

Conclusions

"The robustness and accuracy…27 samples…". The authors should present a table where results of qPCR and biosensor are compared for the 27 spiked samples

"our method does not require further adaptations to accurately detect E. coli…": the authors are somewhat optimistic. They have shown that the method is not totally specific and the biosensor responds not only to DH5alpha but also to TOP10 strain. In addition, other types of strains have not been tested. And the detection has not been evaluated in body fluids. The conclusions should be written with more moderation.

Author Response

Reviewer # 2

Materials and methods:

Comment 1. Lines 89-90 (and fig. 1B) describe the activation of carboxylated electrode surface through EDC/NHS chemistry, followed by chemical binding of antibodies via their amine groups. Some lines after (line 94), and further in the manuscript (lines 159-160), the authors report that activated carboxyl groups of antibodies react with surface amine groups. EDC/NHS chemistry actually activate carboxyl groups which can react with amine groups, not the opposite. Please make the adequate corrections.

Author’s Reply. In the presence of EDC and NHS, the activated groups of thyroxine covalently bonded to the primary amine at the N-terminal (or side-chain amine groups) of antibodies. Please refer to lines 93-94 in the revised manuscript for your consideration.

Comment 2. How are the numbers of bacterial cells determined? Are the different suspension concentrations obtained by successive dilutions of a mother suspension?

Author’s Reply. We used the basic microbiological technique of standard plate count to determine bacterial cells. Number of bacteria per mL =

From a mother stock of 6400 cells, which were determined by standard plate count, we used 1:4 serial dilutions to create decreasing concentrations from the mother sample. Serial plates will be created with a low enough number of bacteria that we can count individual colonies. A range of cells from 25 to 6400 was determined by the same technique and used to load to the chip.

Comment 3. It is mentioned that the specificity studies are performed using the 5 strains at 100 cfu. Results presented in Figure 5A are not consistent with this concentration and Fig 4A.

Author’s Reply: Thank you for your comments. The assays of specificity shown in Fig. 5 A-D were conducted three times and the average value was shown in the updated Fig.5 in the revised manuscript. We carefully vortexed samples 30 seconds to remove cell clusters that could generate local binding onto electrodes, which could be a possible reason for a large decrease in current observed in Fig. 5A in the unrevised manuscript.

Results and discussion

Comment 4. Fig.1B : electron transfer for ferrocene/ferrocenium couple is a one electron process that occurs via iron not benzene rings. Does the ferrocene-tyroxine complex electrochemistry rely on a two-electron process, as suggested in the figure? Please provide a suitable reference that describes the related mechanism.

Author’s Reply. Thyroxine-conjugated ferrocene is capable of functioning as an electron transfer mediator, shuttling electrons between a redox probe [Fe(CN)6]−3/−4 and graphite electrode. Conjugating thyroxine to ferrocene does not affect ferrocene as an electron transfer mediator. In the figure, we use thyroxine as a receptor to conjugate with the antibody using EDC/NHS activation and this use does neither involve nor impair the nature of the electron transfer mechanism of ferrocene (Robinson et al. 1986; Yao and Rechnitz 1987).

Comment 5.  Are there still free iodine groups at the electrode surface (as suggested in the figure and in line 205), or are they all in the Fe complexed form? If any, they will generate repulsion effects (and not attractive interactions, line 206) towards the ferro/ferricyanide anionic probe.

Author’s reply. Thank you for the interesting comment. We opened the discussion since we cannot control the yield of the reaction yielding 100% in which some free functional groups are available. In this scenario, free functional groups include amine (+1) and iodine (-1) (from thyroxine) and ferric (+2) (from ferrocene). The Fe atom of ferrocene readily oxidizes to Fe2+, giving Fe(C5H5)22+ ion when applying a voltage (Bao et al. 2009). The net charge of some sites on electrodes where are not covered by antibodies could be positive charge site for the anionic probe [Fe(CN)6]3−/4 binding. Although, electron transfer to an electrode requires a very close approach, KCl electrolyte support solution can maintain the current as from for the anionic probe [Fe(CN)6]3−/4 to the electrode.

Comment 6. Fig.2C: with the chosen representation (increasing current vs increasing potential), negative peaks (reduction) come on the left part of CV and positive peaks (oxidation) come on the right part of CV. Please correct the figure.

Author’s reply:  Thank you. We corrected the figure according to UIPAC Convention. Please refer to Figure 2C and 2D for your consideration.

Comment 7. Why is Ferro/ferricyanide couple used for detection and not ferrocene- directly? This should be clearly explained. Signals obtained in absence of ferrocene and in presence of ferrocene but in absence of ferro/ferricyanide couple should be given for justification.

Author reply. There are two reasons that we used Ferro/Ferricyanide couple for detection. Although the oxidation of ferrocene to ferricenium ion is a fast reversible electron transfer reaction at most electrodes, ferrocene is water insoluble necessitating the use of a non-aqueous solvent like acetonitrile, not suitable for biosensor if ferrocene in an acetonitrile solution. In this study, we used ferrocene-modified electrode in contact with an electrolyte solution containing potassium ferrocyanide. The reverse reaction of ferricyanide reduction by immobilized ferrocene with 0.1 M KCl electrolyte salt. Figure 2C shows CV of the immobilized ferrocene in acetonitrile and Figure 2D show CV of ferrocene and ferricyanide couple in 0.1 M KCl electrolyte support solution.

Comment 8. Fig.3A and B: The common Bode representation plots logZ vs Logf, not Z vs f. Please, correct also Y axis titles.

Author reply. The Bode curve representation was showed in log(Z) vs log(f). We also corrected Y-axis titles. Please refer to Figure 3A and B in the revised manuscript for your consideration.

Comment 9. Fig.4A: what was the incubation time ? The blank signal (in absence of bacteria) should be also represented.

Author reply. The optimal incubation time was determined approximately in 5 to 7 min due to the highest value of the angular coefficient of the cell–response curves, referring to binding kinetics between the antibody and E. coli.

Comment 10. Fig.4B: equation of the regression line given in the figure is not correct. How are the error bars calculated ? (how many replicates, how are they obtained?). This  curve  (calibration curve) should be represented as delta I vs loc C.

Author reply. The standard error is calculated by dividing the standard deviation by the square root of number of measurements that make up the mean. Here we measured 3 times for the assay, so the standard error is calculated by dividing the standard deviation by the square root of 3. The figure 4B was represented as  ÄI (µA) vs log [cell]. Please refer to Figure 4B in the revised manuscript for your consideration.

Comment 11. Fig.4C: please correct X-axis values and title as well as Y values. How do you explain the large decrease of slope at 2 min?

Author reply. Thank you for this comment. We remeasured the time-dependent assay and found that some errors occurred in the previous experiment such as formation of cell cluster when we leave the diluted samples on ice. The large decrease of slope at 2 min could be from the cell cluster formation. In a new experiment, we vortexed the sample 30 seconds before loading to the sensing chamber. Figure 4A-C were corrected in the revised manuscript, please have suggestions for those corrections.

Comment 12. Line 249: "…approximately in 5 to 7 min". This is not consistent with results presented in Fig4C. The figure or the text should be modified.

Author reply. Figure 4C in the revised manuscript was corrected to be consistent with the text. Please consider figure 4C for your consideration.

Comment 13. Table 1 is not useful, primers and PCR length can be provided in the materials and methods section, directly in the text.

Author reply. Table 1 was removed from the main text. We left the primer sequences in lines 261 -261.

Comment 14. line 262: it can not be affirmed that a regression coefficient of R2= 0.954 is the sign of high linearity.

Author reply. “It is important to highlight that our results (current signal) presented a high linearity (R2 = 0.954) with the Ct values” was deleted. Thank you.

Conclusions

Comment 15. "The robustness and accuracy…27 samples…". The authors should present a table where results of qPCR and biosensor are compared for the 27 spiked samples

Author Reply. We added a table to show comparable results of qPCR and biosensor for the 27 spiked samples.

Table 1. Results of qPCR and biosensor for the 27 spiked samples

Spiking samples

qPCR (Ct values)

Biosensor (µA)

Sample 1

21.5

46

Sample 2

27.8

54.5

Sample 3

19.8

40.2

Sample 4

23.4

47.2

Sample 5

25.7

47.8

Sample 6

26.3

48.2

Sample 7

24.9

47.5

Sample 8

21.8

46

Sample 9

29.3

55.8

Sample 10

20.0

40

Sample 11

27.6

54.3

Sample 12

31.7

57.6

Sample 13

28.9

53.6

Sample 14

21.4

46.2

Sample 15

27.5

53.4

Sample 16

32.8

58

Sample 17

34.3

58.6

Sample 18

19.9

39.5

Sample 19

35.5

60.5

Sample 20

27.2

52.9

Sample 21

26.8

52.7

Sample 22

32.5

58.3

Sample 23

23.7

48.6

Sample 24

28.4

53.1

Sample 25

29.6

53.8

Sample 26

37.3

61.5

Sample 27

26.5

52.4

Comment 16. "our method does not require further adaptations to accurately detect E. coli…": the authors are somewhat optimistic. They have shown that the method is not totally specific and the biosensor responds not only to DH5alpha but also to TOP10 strain. In addition, other types of strains have not been tested. And the detection has not been evaluated in body fluids. The conclusions should be written with more moderation.

Author reply. Thank you. The conclusions were rewritten. The robustness and accuracy of the sensors were successfully evaluated by analyzing 27 spiking samples for whole-cell detection (Table 1), indicating that our method can be an alternative to accurately detect E.coli in spiking samples, but it needs to be challenged to real samples. Additionally, it could enable monitoring other pathogens since the modification of the electrodes with other recognition elements can be readily performed; and it is easy to operate and could integrate to a wireless system. Finally, the sensor can be applied for detecting pathogens in the areas of biosafety, aquaculture, and water quality if the recognition elements (like aptamers and antibodies) are available.

Reference

Bao, D., Millare, B., Xia, W., Steyer, B.G., Gerasimenko, A.A., Ferreira, A., Contreras, A., Vullev, V.I., 2009. Electrochemical Oxidation of Ferrocene: A Strong Dependence on the Concentration of the Supporting Electrolyte for Nonpolar Solvents. The Journal of Physical Chemistry A 113(7), 1259-1267.

Robinson, G.A., Martinazzo, G., Forrest, G.C., 1986. A Homogeneous Bioelectrochemical Immunoassay for Thyroxine. Journal of Immunoassay 7(1-2), 1-15.

Yao, T., Rechnitz, G.A., 1987. Amperometric enzyme-immunosensor based on ferrocene-mediated amplification. Biosensors 3(5), 307-312.

Reviewer 3 Report

Malhotra et al. have fabricated a low-cost template for detection of E. coli. There are several issues that need to be addressed before considering this paper for publication.

  • The use of a 3D printer is not very well justified.
  • There is lack of control in this study, the accuracy and specificity of the proposed biosensor is not compared with conventional/ other detection methods.

Author Response

Review 3. Comments and Suggestions for Authors

Malhotra et al. have fabricated a low-cost template for detection of E. coli. There are several issues that need to be addressed before considering this paper for publication.

Comment 1. The use of a 3D printer is not very well justified.

Author reply. Throughout the past few decades, point-of-care testing (POCT) device have clinically and technologically advanced in the medical discipline, enabling it for both developed and developing countries. Laboratory-based testing poses numerous disadvantages such as, long-time obtainable results, expensive costs, increase chances of contamination, and more. On the other hand, POCT technologies, such as the one presented in this paper, are low-cost, low contamination and shorter time to obtain results, all while allowing for near-patient testing. Even with the beneficiaries of POCT devices, there are restraints in cost, design complexity, and material of current fabrication techniques. Here, we present a POCT device that is based on 3D-printing technology, specifically Fused Deposition Modeling (FDM) that allows it to be rapidly-developed all while being cost-effective. Furthermore, our 3D-Printed POCT device still has the same benefits as current POCT technologies while keeping similar sensitivity to those of competitive platforms.

Comment 2. There is lack of control in this study, the accuracy and specificity of the proposed biosensor is not compared with conventional/ other detection methods. Table provided !

Author reply. Table 2 of comparable parameters between the conventional and biosensor methods is presented. Please refer to Table 2 in the revised manuscript.

Table 2. Examples of characteristics of conventional methods and biosensor for whole cell detections.

Assays

Advantages

Disadvantages

Detection signal

LOD (cfu.mL-1)

PCR

No enrichment steps. Multiple detections

Detection result:  2-5h

No discrimination between viable and non-viable cells 

Electrophoresis

100 – 105 [1]

Based on samples

qPCR

Real time, no post-amplification processing 

Use of fluorescent tags

Fluorescence

1 – 100 [2]

Based on samples

RT-PCR coupled to fluorescence

Fast detection (30 min -1 hour)

Use of fluorescent tags

Fluorescence

102 [3]

Based on samples

Culture and Colony

Particular bacteria species

Excessively time-consuming

Different selective media 

5–14 days

Phenotype Biochemical reactions

1-100 [4]

Based on samples

Immunology-based Methods

A wide range of targets

Time: 2-4 h

Combined with almost any detection method, e.g., optical, magnetic force microscopy, magnetoresistance 

Antibody-antigen interactions

Fluorescent and chromogenic reactions

Aggregations

30-1000 [5]

Based on samples

Optical

biosensor

High selectivity and sensitivity

Label free

Fluorescent probes

Complex system (SPR)

Adds time and cost to the procedure.

Refractive indexes, fluorescent, light scattering

15 – 100 [6]

Based on samples

Square wave voltammetry biosensor

High selectivity and sensitivity

Label free

Time

High probability of cross-sensitivity 

Electrical signal

10–1000 [7, 8]

Based on samples

Impedimetric immunosensors

Non-destructive mechanism

wide range of material,

10 min

High probability of cross-sensitivity 

Electrical signal

3-500 [9, 10]

Based on samples

Thank you.

Reference

  1. Holland, J. L.; Louie, L.; Simor, A. E.; Louie, M., PCR detection of Escherichia coli O157:H7 directly from stools: evaluation of commercial extraction methods for purifying fecal DNA. J Clin Microbiol 2000, 38, (11), 4108-4113.
  2. Kasturi, K. N.; Drgon, T., Real-Time PCR Method for Detection of Salmonella spp. in Environmental Samples. Appl Environ Microbiol 2017, 83, (14), e00644-17.
  3. Hu, L.; Han, B.; Tong, Q.; Xiao, H.; Cao, D., Detection of Eight Respiratory Bacterial Pathogens Based on Multiplex Real-Time PCR with Fluorescence Melting Curve Analysis. Can J Infect Dis Med Microbiol 2020, 2020, 2697230-2697230.
  4. Lagier, J.-C.; Edouard, S.; Pagnier, I.; Mediannikov, O.; Drancourt, M.; Raoult, D., Current and past strategies for bacterial culture in clinical microbiology. Clin Microbiol Rev 2015, 28, (1), 208-236.
  5. Law, J. W.-F.; Ab Mutalib, N.-S.; Chan, K.-G.; Lee, L.-H., Rapid methods for the detection of foodborne bacterial pathogens: principles, applications, advantages and limitations. Frontiers in Microbiology 2015, 5.
  6. Yanase, Y.; Hiragun, T.; Ishii, K.; Kawaguchi, T.; Yanase, T.; Kawai, M.; Sakamoto, K.; Hide, M., Surface plasmon resonance for cell-based clinical diagnosis. Sensors (Basel) 2014, 14, (3), 4948-4959.
  7. Gunasekaran, D.; Gerchman, Y.; Vernick, S., Electrochemical Detection of Waterborne Bacteria Using Bi-Functional Magnetic Nanoparticle Conjugates. Biosensors 2022, 12, (1), 36.
  8. Razmi, N.; Hasanzadeh, M.; Willander, M.; Nur, O., Recent Progress on the Electrochemical Biosensing of Escherichia coli O157:H7: Material and Methods Overview. Biosensors 2020, 10, (5), 54.
  9. Malvano, F.; Pilloton, R.; Albanese, D., Sensitive Detection of Escherichia coli O157:H7 in Food Products by Impedimetric Immunosensors. Sensors (Basel) 2018, 18, (7), 2168.
  10. Leva-Bueno, J.; Peyman, S. A.; Millner, P. A., A review on impedimetric immunosensors for pathogen and biomarker detection. Medical Microbiology and Immunology 2020, 209, (3), 343-362.

Round 2

Reviewer 2 Report

The revised manuscript has been deeply modified according to a range of my remarks. However, there are still some major revisions to be done (see remaining comments below).

Comment 2. How are the numbers of bacterial cells determined? Are the different suspension concentrations obtained by successive dilutions of a mother suspension?

Author’s Reply. We used the basic microbiological technique of standard plate count to determine bacterial cells. Number of bacteria per mL = ?????? ?? ???????? 

From a mother stock of 6400 cells, which were determined by standard plate count, we used 1:4 serial dilutions to create decreasing concentrations from the mother sample. Serial plates will be created with a low enough number of bacteria that we can count individual colonies. A range of cells from 25 to 6400 was determined by the same technique and used to load to the chip.

Reviewer response: Thank you. The way samples are prepared should be mentioned in the manuscript

Comment 3. It is mentioned that the specificity studies are performed using the 4 strains at 100 cfu. Results presented in Figure 5A are not consistent with this concentration and Fig 4A.

Author’s Reply: Thank you for your comments. The assays of specificity shown in Fig. 5 A-D were conducted three times and the average value was shown in the updated Fig.5 in the revised manuscript.

Reviewer response: Thank you for your answer, but I think you misunderstood my remark. Former Fig. 5a and Fig 5b showed strong decrease in signal, from about 65 to 10 microA for DH5, which corresponds to the response to more than 6400 CFU according to Fig.4, and not 100 CFU as indicated. In those conditions, there was also a strong decrease of signal for TOP10 strain (former Fig. 5b), which demonstrates cross-reactivity. You should not modify the figure by showing results for 100 CFU, where deviation is not visible. Please, present former Fig. 5 with adequate comments. Specificity studies should be performed at high CFU values.

Comment 7. Why is Ferro/ferricyanide couple used for detection and not ferrocene- directly? This should be clearly explained. Signals obtained in absence of ferrocene and in presence of ferrocene but in absence of ferro/ferricyanide couple should be given for justification.

Author reply. There are two reasons that we used Ferro/Ferricyanide couple for detection. Although the oxidation of ferrocene to ferricenium ion is a fast reversible electron transfer reaction at most electrodes, ferrocene is water insoluble necessitating the use of a non-aqueous solvent like acetonitrile, not suitable for biosensor if ferrocene in an acetonitrile solution. In this study, we used ferrocene-modified electrode in contact with an electrolyte solution containing potassium ferrocyanide. The reverse reaction of ferricyanide

reduction by immobilized ferrocene with 0.1 M KCl electrolyte salt. Figure 2C shows CV of the immobilized

ferrocene in acetonitrile and Figure 2D show CV of ferrocene and ferricyanide couple in 0.1 M KCl electrolyte support solution.

Reviewer response: I am not sure to understand your answer correctly. What is the CV response of ferrocene at the electrode surface in KCl medium as electrolyte?

There is no legend for Fig 2D

Comment 15. "The robustness and accuracy...27 samples...". The authors should present a table where results of qPCR and biosensor are compared for the 27 spiked samples

Author Reply. We added a table to show comparable results of qPCR and biosensor for the 27 spiked samples.

Reviewer response:  the table should not present intensities of signals. "Spiked samples" means that you added a known concentration of bacteria in one original sample. What is this sample (i.e., what is the sample matrix), what was the added concentrations and what was the concentrations measured by both methods? How many replicate. Measurements have been performed? What is the percentage of deviation between results obtained by your biosensor and the reference method? These results should be included and detailed in the body of the manuscript. Where are robustness results?

Author Response

We appreciate your valuable comments and suggestions. We have carefully revised our manuscript following your comments point-by-point. The revised text was highlighted in red. Your constructive comments are extremely helpful for us to improve our work as well as to better revise the current manuscript.

Below are detailed answers to reviewers’ comments.

Best regards,

Nguyen, et al.

Electrical Engineering and Computer Science

UC Irvine

===

Review 2. Comments and Suggestions for Authors

Comment 2. How are the numbers of bacterial cells determined? Are the different suspension concentrations obtained by successive dilutions of a mother suspension?

Author’s Reply. We used the basic microbiological technique of standard plate count to determine bacterial cells. Number of bacteria per mL = ?????? ?? ???????? 

From a mother stock of 6400 cells, which were determined by standard plate count, we used 1:4 serial dilutions to create decreasing concentrations from the mother sample. Serial plates will be created with a low enough number of bacteria that we can count individual colonies. A range of cells from 25 to 6400 was determined by the same technique and used to load to the chip.

Reviewer response: Thank you. The way samples are prepared should be mentioned in the manuscript

Author’s reply: Thank you, we added the preparation step in the manuscript. Please refer to lines 102 – 107 for your consideration.

The basic technique of microbiology of standard plate count was used to determine bacterial cells, Number of bacteria per mL = . From a mother stock of 6400 cells, which were determined by standard plate count, we used 1:4 serial dilutions to make decreasing from 6400 to 25 colony-forming units (cfu) from the mother sample before loading them to the chip.

Comment 3. It is mentioned that the specificity studies are performed using the 4 strains at 100 cfu. Results presented in Figure 5A are not consistent with this concentration and Fig 4A.

Author’s Reply: Thank you for your comments. The assays of specificity shown in Fig. 5 A-D were conducted three times and the average value was shown in the updated Fig.5 in the revised manuscript.

Reviewer response: Thank you for your answer, but I think you misunderstood my remark. Former Fig. 5a and Fig 5b showed strong decrease in signal, from about 65 to 10 microA for DH5, which corresponds to the response to more than 6400 CFU according to Fig.4, and not 100 CFU as indicated. In those conditions, there was also a strong decrease of signal for TOP10 strain (former Fig. 5b), which demonstrates cross-reactivity. You should not modify the figure by showing results for 100 CFU, where deviation is not visible. Please, present former Fig. 5 with adequate comments. Specificity studies should be performed at high CFU values.

Author reply. We repeated doing those experiments since large variations of batch-to-batch experiments were observed in the former data. Thank you for your useful comments. The former 5a and 5b in the first submission showed a strong decrease in the signal of µA. The reason for the strong decrease (from 60 µA (Fig. 4A) to 10 µA (Fig. 5A) and 21 µA (Fig. 5B) was that we used old bacterial samples that were left in the refrigerator over three days, but we only pipetted the samples before loading to the chip. Leaving bacterial samples in the refrigerator (4oC) over three days could generate layers of cell clusters rather than a single layer of cells in the solution. We illustrate the phenomenon of cells in images below.

The cell clusters on the electrode could make the current strong drop as shown in Fig 5a and Fig 5b.  In a well-distributed format, antibody receptors capture single bacterial cells in a monolayer. In a format of clustered cells, however, antibody receptors capture a group of clustered cells on the electrode, which makes a non-conductive layer on the electrode surface denser. The denser cell layer generates a strong decrease of the current in 65 to 10 µA and to 21 µA as shown in Fig. 5a and Fig. 5b, respectively.

We re-plated and re-run all experiments with fresh bacterial samples aliquoted from the mother stock (6400 cells) and load them to the chip right after. To determine deviations, three samples were measured for each concentration. We added these measurements and their deviations of new Figs. 5A-D in the body of the manuscript. Please refer to lines 369-373 and Fig. 5 in the second-round revised manuscript for your consideration.

Figure 5. Specificity studies of the sensor using four laboratory E. coli strains. (A) DH5α, (B) TOP10, (C) BL21, (D) JM109. Black lines are baseline corrected SWV before incubating with bacteria sample. Red lines for 1 % of BSA solution as an unspecific binding. Green lines for each graph are the result after incubation with the specific E. coli strain. Experimental conditions were conducted as the same conditions described in this study.

Comment 7. Why is Ferro/ferricyanide couple used for detection and not ferrocene- directly? This should be clearly explained. Signals obtained in absence of ferrocene and in presence of ferrocene but in absence of ferro/ferricyanide couple should be given for justification.

Author reply. There are two reasons that we used Ferro/Ferricyanide couple for detection. Although the oxidation of ferrocene to ferricenium ion is a fast reversible electron transfer reaction at most electrodes, ferrocene is water-insoluble necessitating the use of a non-aqueous solvent like acetonitrile, not suitable for biosensor if ferrocene in an acetonitrile solution. In this study, we used a ferrocene-modified electrode in contact with an electrolyte solution containing potassium ferrocyanide. The reverse reaction of ferricyanide reduction by immobilized ferrocene with 0.1 M KCl electrolyte salt. Figure 2C shows the CV of the immobilized ferrocene in acetonitrile and Figure 2D shows the CV of ferrocene and ferricyanide couple in 0.1 M KCl electrolyte support solution.

Reviewer response: I am not sure to understand your answer correctly. What is the CV response of ferrocene at the electrode surface in KCl medium as electrolyte? There is no legend for Fig 2D

Author’s reply. We used a solution containing 200 µL of 10 mM Ferrocene in acetonitrile that was added 2 ml of 100 mM KCl electrolyte-supported medium. The CV response of ferrocene in KCl medium was showed below.

Regarding we used of ferrocene in this assay, the previous study [1] reported that thyroxine has been modified with ferrocene to produce an antibody reactive conjugate which still maintains the potential of functioning as an electron transfer mediator, shuttling electrons between an oxidoreductase enzyme and an electrode. In this experiment, we used ferrocene and thyroxine to respond to two functions that are (i) a conjugation linker and an electron transfer mediator. However, as observed by recorded data, the function of electron transfer mediators is underestimated. In the future, we can design experiments to study and evaluate this function onto surfaces of different electrode materials. We also showed CV of 10 mM of Ferrocene in Acetonitrile in supported electrolyte KCL in Figure 4A of the body of the manuscript. Thank you.

Comment 15. "The robustness and accuracy...27 samples...". The authors should present a table where results of qPCR and biosensor are compared for the 27 spiked samples

Author Reply. We added a table to show comparable results of qPCR and biosensor for the 27 spiked samples.

Reviewer response:  the table should not present intensities of signals. "Spiked samples" means that you added a known concentration of bacteria in one original sample. What is this sample (i.e., what is the sample matrix), what was the added concentrations and what was the concentrations measured by both methods? How many replicate. Measurements have been performed? What is the percentage of deviation between results obtained by your biosensor and the reference method? These results should be included and detailed in the body of the manuscript. Where are robustness results?

Author’s reply. Thank you for your comment.

To prepare spiking samples for both methods, approximately 100 µL of 100 cells were added to each tube containing 400 µL of tape water as sample matrix and vortex in 30 seconds before loading to the chip. Regarding qPCR, 500 µL of each sample of two independent samples were centrifuged at 7,000 rpm for 10 minutes to collect cells for colony qPCR with SYBR master kits (Thermo Fisher). We attached melting curves to see the specific amplification of products.

We also reformatted the table and added a figure (Fig. 6) to show the data of spiking assays. In the last Table 1, we added 100 cells in 27 matrix samples of LB for the assay and presented data were the mean data of the duplicate measurement. In the revision, we performed other measurements for spiked samples in which the sample matrix was tap water, with 1% BSA as a specific control. In this revised format, we presented data in Figure 6(A-C) instead of showing them in tables. Cross-reactivity of the antibody to other E.coli species evaluated through DI was shown in Fig. 6A. Through melting curve analysis, Fig. 6B showed the specificity of the primer pairs for qPCR for DH5alpha and TOP101. Fig 6C showed the correlation regression between biosensing and qPCR method. The data from 27 spiked sample showed that the correlation between the two methods is not high as expected (R2 = 0.78).

Figure 6.  (A) Evaluation of cross-reactive E.coli DH5alpha antibodies to TOP10, BL21 and JM109 species of E.coli. The cut-off value was obtained from the Receiver Operating Characteristics (ROC) Curve (not shown). (B)  Melting curves for specific amplification analysis. (C) Biosensing measurements of E.coli DH5alpha compared with colony qPCR by a regression procedure.

We added those data in the body of manuscript. Please refer to lines 403 – 412 for your consideration.

Thank you for your time and kindness. Your expertise and experiences help us a lot to improve our manuscript and knowledge in this field as well.

Reference

  1. Robinson, G. A.; Martinazzo, G.; Forrest, G. C., A Homogeneous Bioelectrochemical Immunoassay for Thyroxine. Journal of Immunoassay 1986, 7, (1-2), 1-15.

Reviewer 3 Report

The authors have addressed most of the comments mentioned before, the manuscript is suggested for publication in this journal.

Author Response

Thank you for your time and valuable comments.